# The Potential of Calcium/Phosphate Containing MAO Implanted in Bone Tissue Regeneration and Biological Characteristics

**DOI:** 10.3390/ijms22094706

**Published:** 2021-04-29

**Authors:** Shun-Yi Jian, Salim Levent Aktug, Hsuan-Ti Huang, Cheng-Jung Ho, Sung-Yen Lin, Chung-Hwan Chen, Min-Wen Wang, Chun-Chieh Tseng

**Affiliations:** 1Department of Chemical & Materials Engineering, Chung Cheng Institute of Technology, National Defense University, Taoyuan 330, Taiwan; ftvko@yahoo.com.tw; 2System Engineering and Technology Program, National Chiao Tung University, Hsin-Chu 300, Taiwan; 3Department of Materials Science and Engineering, Gebze Technical University, Gebze 41400, Kocaeli, Turkey; salimleventaktug@gmail.com; 4Orthopaedic Research Center, Kaohsiung Medical University, Kaohsiung 80701, Taiwan; hthuang@kmu.edu.tw (H.-T.H.); rick_free@mail2000.com.tw (C.-J.H.); tony8501031@gmail.com (S.-Y.L.); hwan@kmu.edu.tw (C.-H.C.); 5Department of Orthopedics, Kaohsiung Medical University Hospital, Kaohsiung Medical University, Kaohsiung 80701, Taiwan; 6Regeneration Medicine and Cell Therapy Research Center, Kaohsiung Medical University, Kaohsiung 80701, Taiwan; 7Departments of Orthopedics, College of Medicine, Kaohsiung Medical University, Kaohsiung 80701, Taiwan; 8Department of Orthopedics, Kaohsiung Municipal Ta-Tung Hospital, Kaohsiung 80145, Taiwan; 9Musculoskeletal Regeneration Research Center, Kaohsiung Medical University, Kaohsiung 80701, Taiwan; 10Department of Healthcare Administration and Medical Informatics, Kaohsiung Medical University, Kaohsiung 80701, Taiwan; 11Institute of Medical Science and Technology, National Sun Yat-Sen University, Kaohsiung 80424, Taiwan; 12Department of Mechanical Engineering, National Kaohsiung University of Science and Technology, Kaohsiung 807, Taiwan; mwwang@nkust.edu.tw; 13Combination Medical Device Technology Division, Medical Devices R&D Service Department, Metal Industries Research & Development Centre, Kaohsiung 802, Taiwan

**Keywords:** magnesium alloy, micro-arc oxidation (MAO), SBF, EDTA, biodegradability

## Abstract

Micro arc oxidation (MAO) is a prominent surface treatment to form bioceramic coating layers with beneficial physical, chemical, and biological properties on the metal substrates for biomaterial applications. In this study, MAO treatment has been performed to modify the surface characteristics of AZ31 Mg alloy to enhance the biocompatibility and corrosion resistance for implant applications by using an electrolytic mixture of Ca_3_(PO_4_)_2_ and C_10_H_16_N_2_O_8_ (EDTA) in the solutions. For this purpose, the calcium phosphate (Ca-P) containing thin film was successfully fabricated on the surface of the implant material. After in-vivo implantation into the rabbit bone for four weeks, the apparent growth of soft tissues and bone healing effects have been documented. The morphology, microstructure, chemical composition, and phase structures of the coating were identified by SEM, XPS, and XRD. The corrosion resistance of the coating was analyzed by polarization and salt spray test. The coatings consist of Ca-P compounds continuously have proliferation activity and show better corrosion resistance and lower roughness in comparison to mere MAO coated AZ31. The corrosion current density decreased to approximately 2.81 × 10^−7^ A/cm^2^ and roughness was reduced to 0.622 μm. Thus, based on the results, it was anticipated that the development of degradable materials and implants would be feasible using this method. This study aims to fabricate MAO coatings for orthopedic magnesium implants that can enhance bioactivity, biocompatibility, and prevent additional surgery and implant-related infections to be used in clinical applications.

## 1. Introduction

Metal and its alloy are widely used in bioengineering applications as scaffolds, bone screws, bone plates, and tissue repair and regeneration within the human body; magnesium (Mg) and its numerous alloys are the one of most popular biometal among others [1,2,3]. Because of the excellent biodegradability properties, magnesium alloys have been rapidly developed and widely studied [4,5,6,7].

In recent years, magnesium alloys are developed as biodegradable medical materials. Magnesium is an essential element in the body and is effectively excreted by the kidneys. In patients with bone implants, elastic modulus matching is a crucial consideration in the choice of metallic implant materials for bone-fracture healing because a potential mismatch leads to stress shielding and consequent osteopenia. Another advantage of the biodegradable properties of Mg alloys also avoid a second surgery for implant removal, which prevents additional medical therapy such as inflammation, increased sensitivity to pain and extra costs to the patients [8]. Most traditional metallic materials stay as an anchor in the body for a longer duration of time, leading to corrosion anomalies due to the degradation of the implanted product [9]. In the case of stents, contemporary stents are manufactured from stainless steel (316L), which has high corrosion resistance and remains as a permanent implant in the body. However, this approach has many limitations, including chronic irritation and the release of toxic substances. In contrast, Mg alloy stents are absorbed into the body, thereby overcoming many of these limitations and simplifying secondary surgery following vessel re-occlusion. In the worst cases, large hydrogen bubbles in the circulatory system may block the bloodstream and cause death [6]. However, owing to high electrochemical corrosion, the poor corrosion resistance of magnesium alloy hinders their use in clinical applications. Thus, surface treatment is indispensable to improve its initial corrosion resistance and biocompatibility. Therefore, strategies for controlling the corrosion rates of Mg alloys have been demonstrated in several studies [10,11,12,13].

In most previous research related to the quantitative analysis of Mg alloy corrosion, the researchers focused on overall changes measured using mass loss rate (MLR, mg/mm^2^h) and electrochemical techniques on their specimens [14,15,16]. However, these methods cannot be used to analyze the regeneration of bone tissue in vivo. Changes in the overall corrosion of the Mg alloy do not provide sufficient information to determine whether the mechanical properties of the implant are adequate to maintain its function. Structure analysis of the implant provides better insight into the alloy’s performance. Then, metals for bone implants are widely used in orthopedic surgery. To reach the criteria of metals for bone implants substituted for original damaged bones, researchers designed various bone implants based on different parts of human bone, such as knee joints [17,18], hip joints [19], tooth implants [20,21], and spine implants [22,23,24]. According to different implants, we need to consider biocompatibility, mechanical property, corrosion/ wear resistance, and osseointegration of bone metal implants to compare with the original bone tissue.

Among the various alloplastic materials currently available for the reconstruction of bone defects, Ca-P ceramics possess the most attractive biological profiles. Generic properties of these ceramics include the lack of local or systemic toxicity, no apparent retardation of normal bone mineralization process and rapid appositional bone growth when placed in an osseous bed. As a chemical class, Ca-P ceramics are well tolerated by soft tissue and do not elicit a significant inflammatory or foreign body response. Cultured human fibroblasts have been demonstrated to attach readily to the surface of Ca-P ceramics [25]. Although devoid of any demonstrable osteogenic effects, the chemical nature of these Ca-P materials is believed to facilitate osseous integration [26]. Jahn [27] has suggested that hydroxyapatite (HA) acts as an amphoteric ion exchanger. Selective accumulation of calcium and phosphate ions occurs as a consequence of the negative charges on the HA surface. This leads to the formation of more apatite and stimulates the formation of the new bone substructure. The excellent bonding between new bone substructure and Ca-P implants has been characterized in detail by various investigators [28,29].

According to previous studies [30], it has successfully been designed a micro arc oxidation (MAO) coating with good anti-corrosion performance under pulsed bipolar DC current conditions in a Si- and P-containing electrolyte. The MAO treatment is applied by modifying the surface properties of AZ31 Mg alloy to improve the biocompatibility of the test material in rabbit implant application. The Ca-P has deposited on AZ31 substrate by using an electrolytic mixture of Ca_3_(PO_4_)_2_ and C_10_H_16_N_2_O_8_ (ethylenediaminetetraacetic, EDTA) solutions. Applying the additives of calcium and phosphate with high concentration changes the microstructure of the coatings and produced a less porous and denser MAO structure which improves the corrosion resistance and biocompatibility. In this study, the Ca-P containing MAO samples were immersed in simulated body fluid (SBF) to improve coating surface properties for advanced medical applications. The phase structure and morphology of those combined coatings were characterized, and their biocompatibility was assessed by in vitro cytotoxicity test, histological analyses and animal experiments. Therefore, this study aims to fabricate bioactive and biocompatible Ca-P components on the AZ31 Mg alloy surface.

## 2. Experimental

### 2.1. Preparation of Specimens

Plates of die-cast AZ31 magnesium alloy (3.17 wt % Al, 0.81 wt % Zn, 0.334 wt % Mn, 0.005 wt % Fe, 0.02 wt % Ni, 0.005 wt % Cu, and Mg balance) with the dimension of 50 × 50 × 2.0 mm were used as metallic substrates. Prior to MAO treatment, all plates were mechanically ground by SiC papers up to 2000 grit to ensure the same surface roughness. The plates were then ultrasonically cleaned in acetone, rinsed with deionized water and dried in a hot air stream of 60 °C. The process to prepare MAO coatings on the AZ31 magnesium surface was carried out on a pulse power generator (MIRDC) with work voltage up to 400 V, current up to 10 A, duty cycle at 20% and the electrical frequency at 4000 Hz for 30 min [30]. The water bath made of stainless steel has 45 L water capacity. A glass beaker consists of 1 L of electrolyte was placed inside the water bath. A stainless-steel plate was placed inside the glass to serve as a cathode and the AZ31 alloy substrate was used as an anode in the process. Table 1 shows the bath composition for MAO treatment, in which the entire treatment procedure by the recirculation of cold water was maintained at 25 °C.

### 2.2. Characterization

Microstructure images of the MAO coatings specimen both plane surface and cross-sectional surface were obtained using scanning electron microscope (SEM, JEOL JSM-IT100). Image J software was processed from the stored SEM images for the quantification of pore characteristics (number and average size of pores). In addition, the crystal structure with the crystalline film on a substrate was characterized by X-ray diffraction (XRD; Philips powder X-ray diffractometer PW 1710) with Cu Kα radiation (wavelength 0.15405 nm). The roughness in the coating was investigated via the ET400A (α-Step Talysurf, Sutronic 3+ profilometer). The average roughness (R_a_) was used in this study. In the hydrogen evolution method, the amount of dissolved magnesium was calculated from the volume of hydrogen evolution as a result of the corrosion reaction [31,32]. The PHI 5000 VersaProbe X-ray photoemission spectroscopy (XPS) was used to analyze the surface compositions of the MAO films. Additionally, passive layer structure on the surface was analyzed after immersed in SBF for 2 days. As for XPS spectra, all binding energies were calibrated through C 1s peak at 284.8 eV.

### 2.3. Electrochemical Measurements and Corrosion Test

The electrochemical behavior of the substrate, its bio-corrosion properties and MAO coated samples in SBF (Hank’s solution, pH: 6.5) at 37 °C are tested by potentiodynamic polarization test which are conducted by Autolab PGSTAT30 potentiostat-frequency analyzer. A standard three-electrode system is used. A saturated calomel electrode is used as reference electrode, and all potentials are expressed with respect to this electrode. The platinum plate was used as the counter electrode and specimens were used to be a working electrode with an immersion area of about 1 cm^2^. The state of electrochemical surrounding with specimens has to be steady in SBF until the open-circuit potential (OCP) changed by no more than 2 mV/10 min before the potentiodynamic polarization measurements. After stabilization of 30 min at the OCP, the potential scan rate was controlled at 0.5 mV/sec from −300 mV to 500 mV based on the OCP. The corrosion current density (*i_corr_*), corrosion potential (*E_corr_*), and corrosion rate can be determined by the Tafel extrapolation method from the anodic polarization plots. The salt spray test (SST) was performed for each coated AZ31 plate placed at a tilted angle of 30° in a chamber containing 5 wt% NaCl fog as described in ASTM standard B117. After the salt spray test, the percentage of pitting area was examined by the ASTM D610-08 standard. All the tests were repeated three times.

### 2.4. Mechanical Testing

The bone screw was manufactured to the tensile specimen followed by the design, shown in Figure 1a. The sample was fixed in a custom-made testing jig to confirm pure axial shear loading of the locking screws, as shown in Figure 1b,c. Screw into the D3 bionic bone (Figure 1c) with a force of 3.5 kg. The bone screw was locked and fixed into D3 bionic bone about 20 mm, according to ASTM F543 standard. Then, pull up at a speed of 5 mm/min until the bone screw is pulled out of the D3 bionic bone (Figure 1d). The mechanical properties were measured by a biaxial servo hydraulic machine (MTS Mini Bionix II 858). In summary, the value of each electrochemical corrosion test, surface roughness measurement, porosity analysis and mechanical test were described as the average ± of the standard deviation of the three measurements.

### 2.5. Animal Surgery and Implant Harvest

In vitro cytotoxicity test was performed in this study to evaluate the biological compatibility of MAO coated samples. Extraction of test sample and treatment of murine lung fibroblast cells (L929 cells) with test sample extracts was performed according to ISO10993-12 and ISO10993-5, respectively. Cell viability determined by MTT assay showed that the test sample extract had in average < 30% inhibitory effects on the viability of cells, which was examined by Société Générale de Surveillance (SGS) Taiwan Ltd.

Animal experiments were conducted under the National Institute of Health (NIH) guide for the care and use of laboratory animals and approved by the animal ethics committee of Kaohsiung Medical University (NO: IACUC-103052). New Zealand white rabbits (3.5–4.5 kg, Livestock Research Institute, Taiwan) were used. All animals were kept in a single room and fed with dried diet and water ad libitum, anesthetized with subcutaneous injection of ketamine 40 mg/kg and xylazine 10 mg/kg and then the MAO coated AZ31 screw samples were implanted into the femoral shaft of a rabbit. At 4, 8, and 12 weeks post-implantation, rabbits were euthanized humanely with an intravenous overdose of barbiturate (200 mg/kg).

In order to visualize the samples and analyze images of new bone formation three-dimensionally, the samples were scanned using a micro-computed tomography scanner (µCT, Skyscan 1272, Bruker, Kontich, Belgium) with a high resolution in vivo µCT scanner for preclinical research. A frame averaging of 3 was employed together with a filter of 0.11 × 2 mm copper. The X-ray tube voltage was 100 kV, exposure time of 2050 ms and the current was 100 A. The compiled CT films were viewed and analyzed using NRecon software where a 3-D model is built to determine the quality of bone regeneration. After the implantation, the rabbits were housing in cages individually and monitored by an experienced veterinarian for signs of infection, inflammation and any adverse reaction. The skin was dissected, the implantation site and surrounding bone was harvested by mini-saw. The specimens were fixed in 10% buffered neutralized formalin for 24 h at room temperature and prepared for µCT and histological analyses.

## 3. Results and Discussion

### 3.1. Specimen Surface Characteristic Analysis

#### 3.1.1. Surface Morphology and Chemical Composition

The SEM morphology of the Ca-P containing MAO samples with typical pancake-like structure (a cratered structure with a central pore), as shown in Figure 2. Compare with our previous study [30], the Ca-P containing MAO samples showed a relatively uniform and smooth surface than the Ca-P-free MAO samples. Calcium phosphate had very low solubility in water [33] because the individual calcium cations and phosphate anions were tightly bonded to each other in an aqueous medium. However, by the aid of acid addition such as EDTA, solubility of calcium phosphate was increased in aqueous solution [34,35,36,37]. Then, the pores size and pores number decreased with increasing the concentration of Ca-P containing MAO treated samples. It was also reported that the addition of EDTA as electrolyte component can be helpful to reduce the coating defects such as cracks and pores formed in PEO coating layer [38,39]. Similarly, the surface distribution of the pores was more uniform with increasing the concentration of Ca-P containing MAO samples. In other words, after Ca-P containing coatings by MAO treatment on AZ31 Mg alloys, the surface morphologies were smoother, covered by Ca-P containing thin film layer.

The EDS analysis of the different Ca-P containing MAO samples on AZ31 Mg alloys, as shown in Table 2. The EDS spectrum of the MAO specimen contained magnesium, aluminum, oxygen, silicon, phosphorus, and calcium. As the concentration of EDTA increased, the proportion of Ca in the MAO coating increased. Compared with the SEM surface morphology, when the concentration of the Ca in the MAO coatings decreased, the holes became larger and distributed nonhomogeneously all over the surface.

#### 3.1.2. Cross-Section SEM Characterization

Figure 3 illustrates SEM morphologies of the cross-section and EDS line profile of the Ca-P containing MAO samples. The thickness increases with increasing the concentration of Ca-P containing MAO treated. Kyrylenko et al. [39] reported that EDTA improves the efficiency of accumulating calcium and phosphorus ions into the MAO surface layers. This reaction occurred faster if the addition of EDTA was increased because there are more acid molecules to pull apart the calcium and phosphate from the main substance. It was shown that thickness is increasing from 7.52 μm to 10.49 μm. Thus, it is concluded that the solid calcium phosphate dissolves and provides more Ca^+2^ and PO_4_^−3^ ions into the solution. It should be noted that we can obtain a quite thick coating layer and uniform thickness which is about 10–11 μm at high concentration of Ca-P containing MAO treated. Accordingly, the proportion increases of Ca-P containing could uniformly deposit on AZ31 Mg alloys.

#### 3.1.3. Surface Phase Composition Analysis

Figure 4 showed the XRD patterns of the Ca-P containing MAO samples. XRD patterns showed the crystal structure of the specimen surfaces. The results indicate that the Ca-P containing MAO coatings are mainly composed of MgSiO_3_, MgSiO_4_, and MgO crystallin phases regardless of the different electrolyte compositions. It is also noted that Ca and P related components were not detected in the XRD analysis. However, according to EDS analysis, Ca-P coatings were determined on the AZ31 substrates during the MAO. This is attributed to the existence of amorphous structure of Ca-P related components in the MAO treated coatings that will be revealed later in the XPS analysis.

#### 3.1.4. Electrochemical Measurements and Corrosion Test

Figure 5 shows the results of the 24-h SST for the Ca-P containing MAO samples. There are no rust spots on the different test samples, which mean that not affect the corrosion resistance for the additives of calcium and phosphate in the MAO treatment. To confirm that the additives of calcium and phosphate whether increase the corrosion resistance of the MAO coating, the corrosion behavior of the Ca-P containing MAO samples is determined using the potentiodynamic polarization measurement and hydrogen evolution method.

Figure 6 illustrated the polarization curves of the Ca-P containing MAO samples in SBF solution. The Ca-P free MAO sample showed a corrosion potential (*E_corr_*) of −1.70 V vs. AgCl and a corrosion current density (*i_corr_*) of 8.05 × 10^–7^ A/cm^2^ [30]. The *i_corr_* of the Ca-P containing MAO samples under the amount of EDTA at 2.5, 5, and 7.5 g are 4.04 × 10^−7^ A/cm^2^, 3.45 × 10^−7^ A/cm^2^, and 2.81 × 10^−7^ A/cm^2^, respectively. Hence, the *i_corr_* decreases with increasing the concentration of Ca-P containing MAO treated. All Ca-P containing MAO samples show a lower corrosion current density than that of the Ca-P free MAO sample. On the other hand, the corrosion current density decreases almost one order of magnitude for the Ca-P containing MAO sample compared to the Ca-P free MAO sample.

Figure 7 and Table 3 show the records of the cyclic hydrogen evolution of the Ca-P containing MAO samples and the calculated annual corrosion rates in the SBF solution. The annual corrosion rates of the Ca-P containing MAO samples under the amount of EDTA at 2.5, 5, and 7.5 g are 1.79 ± 0.38 mm/y, 1.47 ± 0.29 mm/y, and 1.30 ± 0.21 mm/y, respectively. Similarly, the corrosion rates decrease with increasing the concentration of Ca-P containing MAO treated, which is consistent with the potentiodynamic polarization measurement. In a word, increasing the concentration of calcium and phosphate can effectively reduce the corrosion rate to protect the AZ31 Mg alloy. Mg alloys are attractive biodegradable medical materials, but their corrosion is accompanied by hydrogen evolution, which is detrimental to the surrounding tissue [40,41,42].

#### 3.1.5. Surface Roughness and Porosity

The average surface roughness of the Ca-P containing MAO samples is also listed in Table 3. The values of the average surface roughness of the Ca-P containing MAO samples under the amount of EDTA at 2.5, 5, and 7.5 g are 0.85 ± 0.19 μm, 0.79 ± 0.14 μm, and 0.62 ± 0.10 μm, respectively. The average surface roughness decreases with increasing the concentration of Ca-P containing MAO treated. When the additives of calcium and phosphate increase to a certain proportion, the MAO coating will gradually become uniform and smooth, and the roughness starts to decrease to a certain level.

Table 3 shows the surface porosity analysis of the Ca-P containing MAO samples, which were calculated by Image-J software for the micro-pores quantification. The porosity of the Ca-P containing MAO samples under the amount of EDTA at 2.5, 5, and 7.5 g are 2.59 ± 0.47%, 2.23 ± 0.45%, and 2.08 ± 0.45%, respectively. Therefore, at the same MAO treatment, applying the additives of calcium and phosphate with high amount changes the microstructure of the coatings, becoming a less porosity and thicker MAO film, which improve the corrosion resistance.

#### 3.1.6. Mechanical Properties

In summary, the AZ31 magnesium alloy bone screw (5 × 30 mm) was made with the best parameters (the Ca-P containing MAO samples under the amount of EDTA at 7.5 g), and the bone screw locking force test was performed. Screw the non-immersed SBF and the bone screws immersed for 6 and 10 weeks into the pre-drilled and tapped imitation bone with a force of 3.5 kg. Then, lock it into the imitation bone 20 mm with ASTM F543 standard and fix it. Finally, stretch upward at a speed of 5 mm/min until the bone screw is pulled out of the imitation bone. The results show that after 10 weeks of immersion in SBF, the AZ31 Mg alloy bone screw retains 86% (265 ± 42 N) of the original locking force (308 ± 40 N), as shown in Table 4.

### 3.2. Biocompatibility Test

#### 3.2.1. In Vitro Cytotoxicity Test-MTT Assay

According to the nature and duration of the anticipated contact with human tissues when in use, medical devices should be carefully tested for biocompatibility to avoid potential physiological damage by toxic substances produced or contaminated during manufacturing. In this study, the Ca-P containing MAO sample was subjected to in vitro cytotoxicity test to evaluate the toxicity of substances that could be extracted or released from the medical device. Based on recommendations described in ISO10993-5, quantitative determination of cell viability by MTT assay and qualitative observation of cell morphology were carried out, followed by concluding level of cytotoxicity according to the scoring criteria listed in the document, which distinguished the test sample and the control sample.

The extracts of the test sample were no different from the blank control, as shown in Figure 8. Table 5 and Figure 8 show the cells exposed to negative control showed no significant change in cell morphology compared to that of reagent control and resulted in a score as 0. Positive control extract caused severe cellular damage and obvious morphological alteration in almost all cells. Therefore, the positive control experiment was scored as 4. The cells treated with the test sample extract showed discrete intracytoplasmic granules and no cell lysis. Therefore, the cell morphology was scored as 0. The acquired readings of OD_570_ absorbance of reagent control were averaged and set as 0% inhibition of cell viability. In proportion to reagent control, we determined inhibition of cell viability of negative control, positive control, and test sample as <30%, 98.2%, and <30%, respectively. The relative values of the inhibition of cell viability were shown in Table 5. The scores of the morphological evaluation and the relative inhibition of cell viability were averaged and listed in Table 5. Based on the ISO10993-5 guidelines, the Ca-P containing MAO sample extract did not induce cytotoxic to L929 cells.

#### 3.2.2. XPS Analysis

To investigate surface chemical compositions of MAO coated samples, we carried out X-ray photoelectron spectroscopy (XPS), as shown in Figure 9. Figure 9a shows the XPS surface full spectrum of the AZ31 sample treated in the EDTA 0 g bath that five elements are detected (Mg, O, P, Al, and Si). Relatively, the XPS surface full spectrum of the AZ31 sample treated in the EDTA 7.5 g bath that six elements are detected (Mg, O, P, Al, Si, and Ca), as shown in Figure 9c. As the EDTA concentration increases in the solution, the solubility of calcium phosphate increases in parallel and this means that the amount of calcium and phosphate ions in the solution increases. Therefore, it illustrates that additives of EDTA can effectively deposit Ca ions on the MAO coating [43]. The atomic percentage of elements in the MAO coating in Figure 9b,d show the XPS depth profile of the MAO coated samples, which exhibit uniform distribution with respect to depth. On the original surface of the MAO coating (about 10 nm; blue box mark in Figure 9b,d), presented the content of Ca and P elements and list in Table 6.

To understand biocompatibility and surface chemical compositions, the MAO coated samples immersed in the SBF solution for 48 h and then carried out XPS analysis as shown in Figure 10. After 48 h immersion in the SBF solution, both samples are detected six elements (Mg, O, P, Al, Si, and Ca). Similarly, on the original surface of the MAO coating in Figure 10a,c (about 10 nm; blue box mark in Figure 10b,d), identifies Ca and P elements and also listed in Table 6. On the other hand, the signals of Si in the AZ31 sample treated in the EDTA 7.5 g bath are detected at a deeper position than the AZ31 sample treated in the EDTA 0 g bath, indicating that the coating with a higher Ca content has better biocompatibility and better corrosion resistance. XPS measurements showed that Ca and P containing components were effectively formed on the surface of the coatings immersed in SBF media after MAO treatment, as shown in Figure 10b,d.

### 3.3. Animal Experiments

#### 3.3.1. Radiological Examination

X-ray findings are taken to reveal the changes that has been undergone. Plain X-rays are the simplest medical images created through X-radiation. Plain X-ray and the implantation films are indicators of tissue changes from pre-surgery to 3 weeks after surgery in rough scale, as shown in Figure 11. Healing of bones on and around the defects can be seen 3 weeks after the operation. The bone screws were no abnormality detected and the bone plates in all subjects were found to be fixed to the bone rigidly and in the original position. Besides, there is no air noted around screws indicating that there were no rapid degradation of screws in the femoral shaft of rabbits.

#### 3.3.2. μCT Scanning

In our previous study [30], bare AZ31 degraded quickly, whereas Ca-P free MAO screw showed a slow degradation rate. Ca-P free MAO screw maintained the original morphology within 12 weeks, whereas AZ31 screw (uncoated) suffered obvious corrosion that several pits present on the surface were observed within 4 weeks in the femoral shaft of a rabbit drill. Figure 12 shows the μCT sequels revealed the preliminary results of the bio-degradation of the AZ31 screw treated in the EDTA 7.5 g bath after implan-tation of (a) 4, (b) 8, and (c) 12 weeks. The difference from the previous study is some new growing bone developed directly on the surface of the AZ31 screw and its continuity was established as bone cell tissue there. From the results of this study, the Ca-P containing MAO screw is postulated to be a potentially promising material in clinical practice as regards to its ability to reinforce the fixation of the bone tissue attaching and to augment the overall effectiveness of tissue healing in bone.

## 4. Conclusions

This research includes an emerging innovative technique for the study of the interface between bone/biomaterials. In summary, the main findings of the work are:

After 10 weeks of immersion in SBF, the AZ31 Mg alloy bone screw retains 86% (265 N) of the original locking force (308 N).

Based on the ISO10993-5 guidelines, the Ca-P containing MAO sample extract did not induce cytotoxicity to L929 cells.

The addition of EDTA into the electrolyte solution with high concentrations of Ca and P has altered the microstructure of the coatings and formed less porous and thicker MAO film that enhances the corrosion resistance and biocompatibility.

MAO can prevent rapid degradation of AZ31 Mg alloy bone screw both in vitro and in vivo.

## Figures and Tables

**Figure 1 ijms-22-04706-f001:**
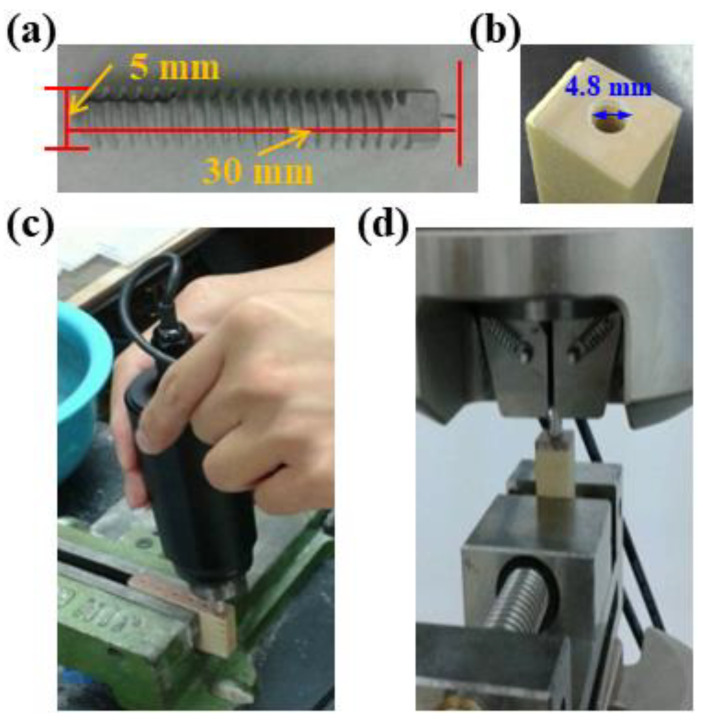
Test setup for the locking force (**a**) bone screw, (**b**) D3 bionic bone, (**c**) custom-made testing, (**d**) pull up testing of the AZ31 Mg alloy bone screw

**Figure 2 ijms-22-04706-f002:**
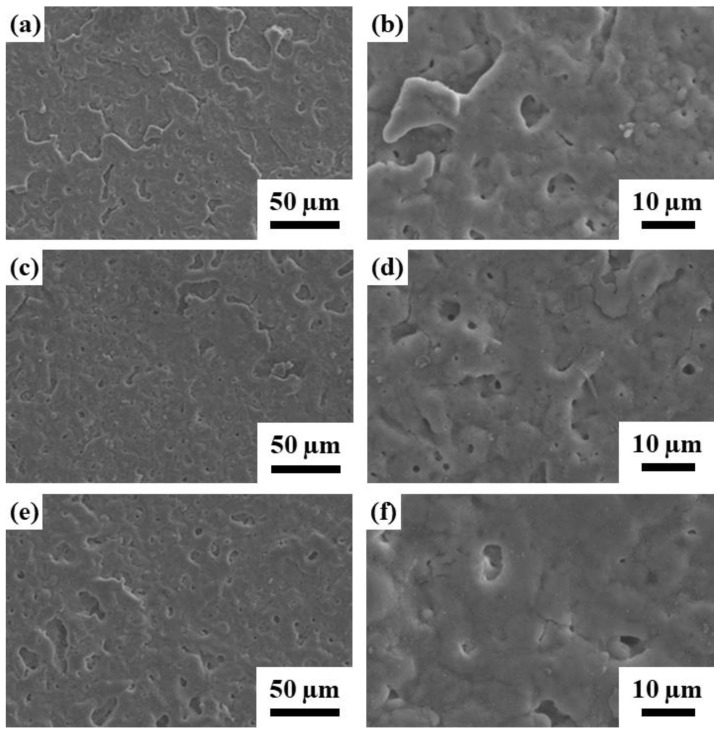
SEM morphology of the AZ31 sample treated in the EDTA (**a**,**b**) 2.5 g; (**c**,**d**) 5.0 g; and (**e**,**f**) 7.5 g bath.

**Figure 3 ijms-22-04706-f003:**
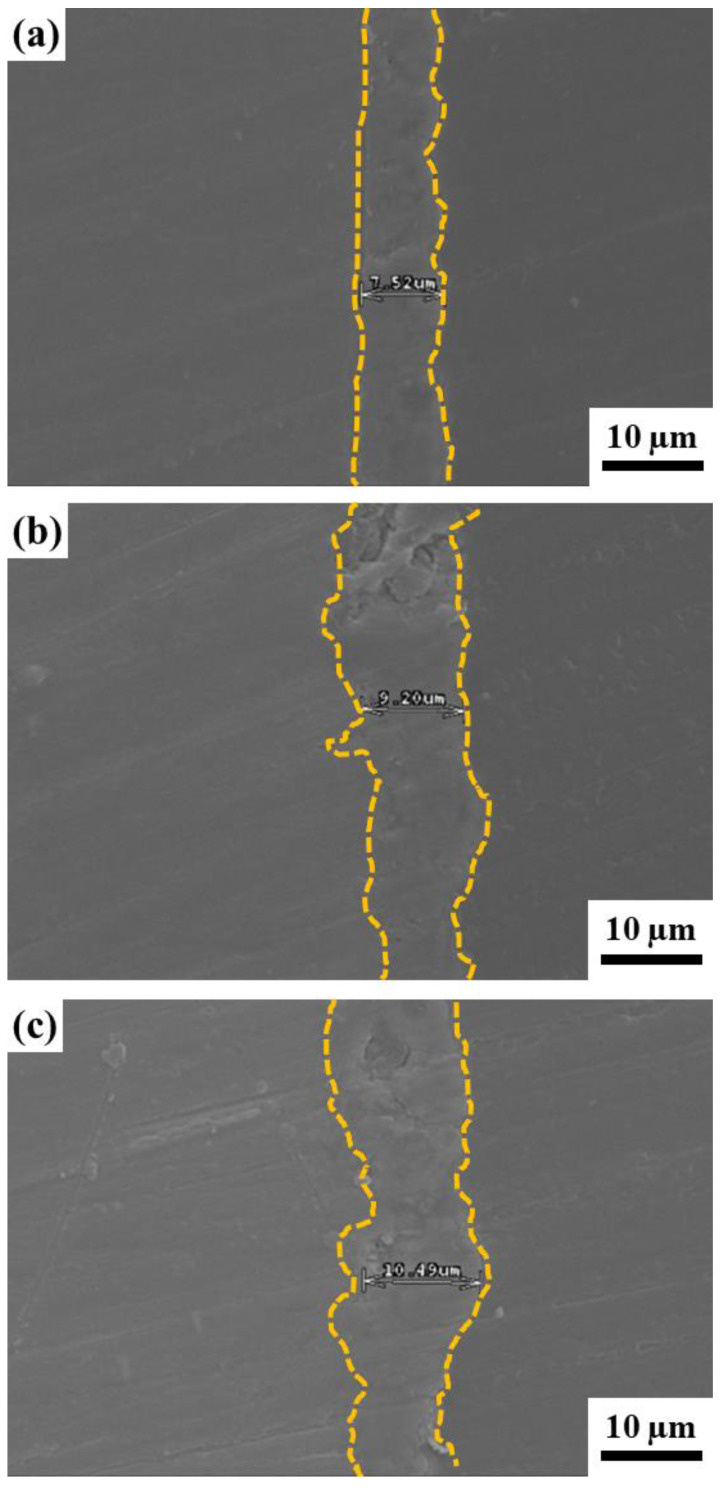
Cross-section SEM morphology of the AZ31 sample treated in the EDTA (**a**) 2.5 g, (**b**) 5.0 g, and (**c**) 7.5 g bath.

**Figure 4 ijms-22-04706-f004:**
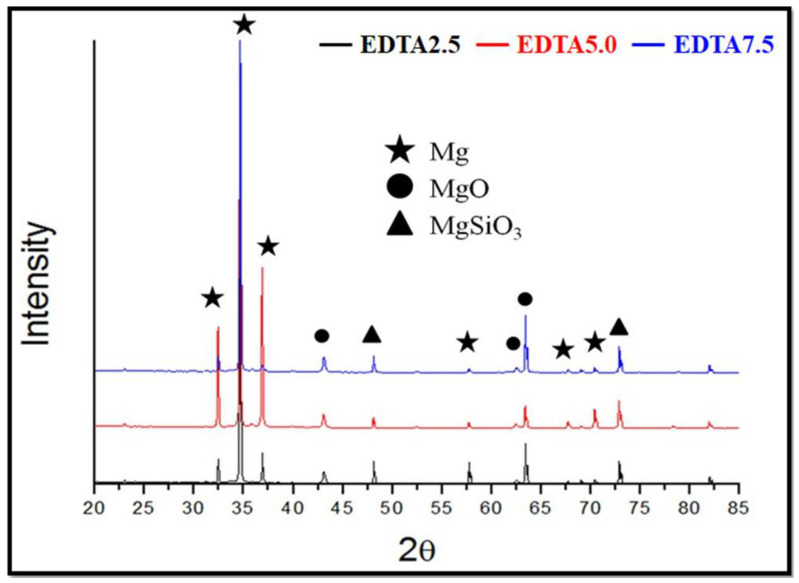
XRD patterns of the AZ31 sample treated in the EDTA 2.5, 5.0, and 7.5 g bath.

**Figure 5 ijms-22-04706-f005:**
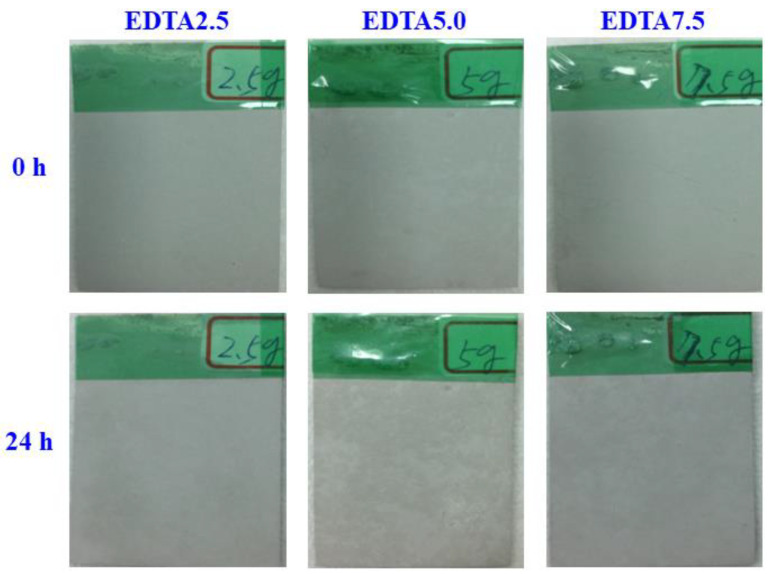
Visual images of the Ca-P containing MAO samples treated in the EDTA 2.5, 5.0, and 7.5 g bath after 24 h of the salt spray test.

**Figure 6 ijms-22-04706-f006:**
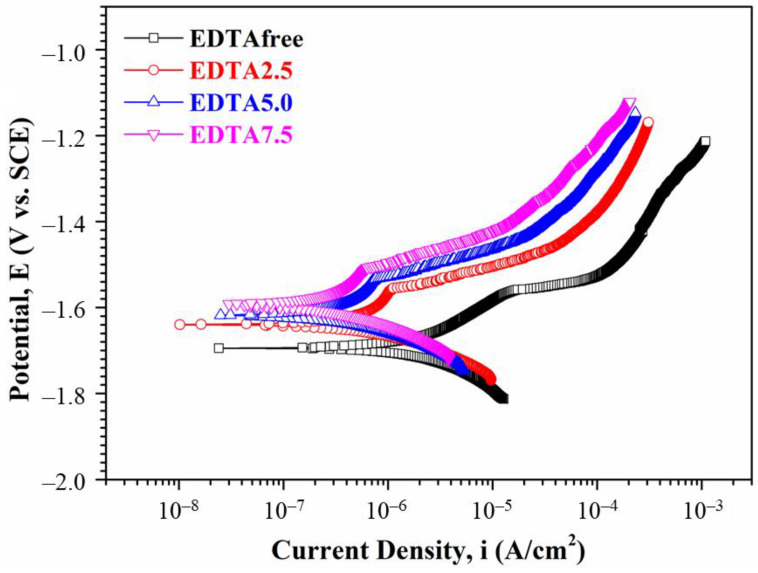
Potentiodynamic polarization curves in SBF solution for the AZ31 sample treated in the various EDTA bath.

**Figure 7 ijms-22-04706-f007:**
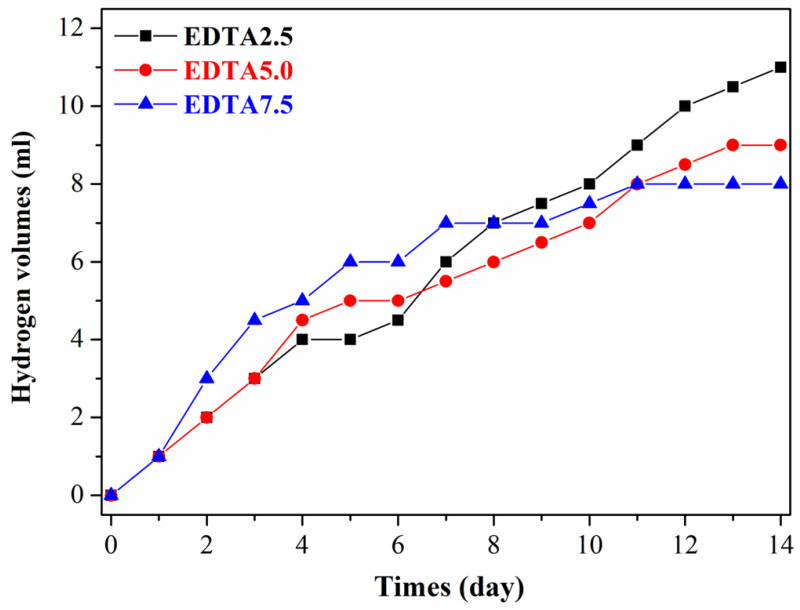
Records of the cyclic hydrogen evolution in SBF solution for the AZ31 sample treated in the various EDTA bath.

**Figure 8 ijms-22-04706-f008:**
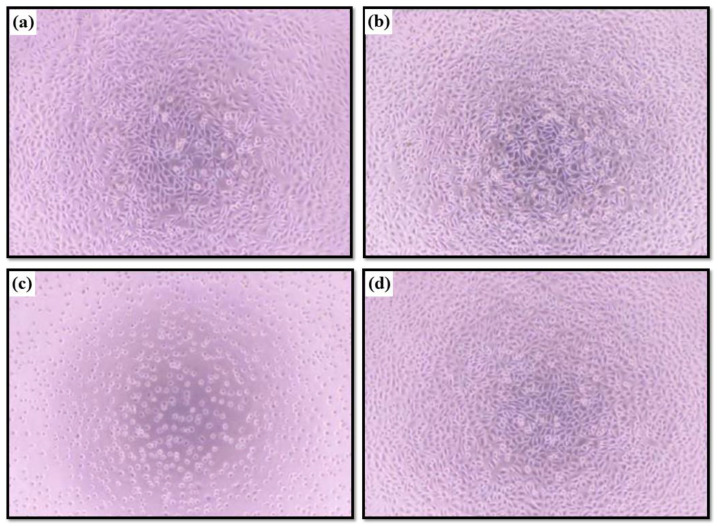
Morphology and confluence of L929 cells after being exposed to (**a**) the Ca-P containing MAO sample, (**b**) blank control, (**c**) positive control, and (**d**) negative control extracts.

**Figure 9 ijms-22-04706-f009:**
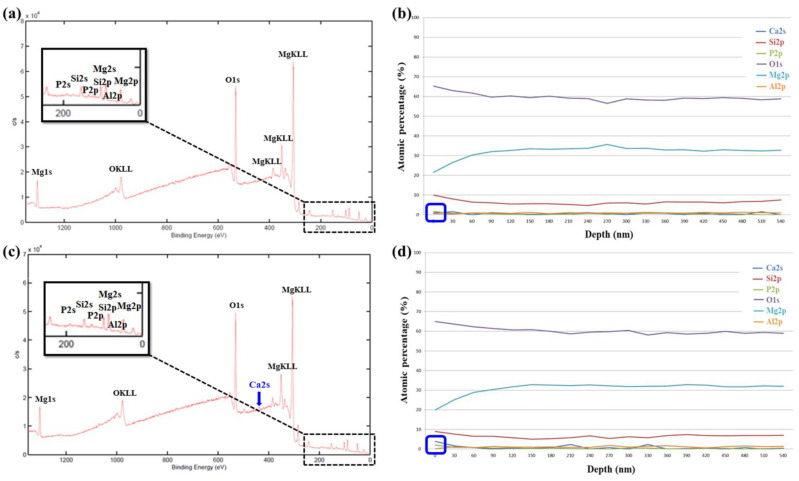
XPS surface full spectrum and depth profile of the AZ31 sample treated in the EDTA (**a**,**b**) 0 g and (**c**,**d**) 7.5 g bath.

**Figure 10 ijms-22-04706-f010:**
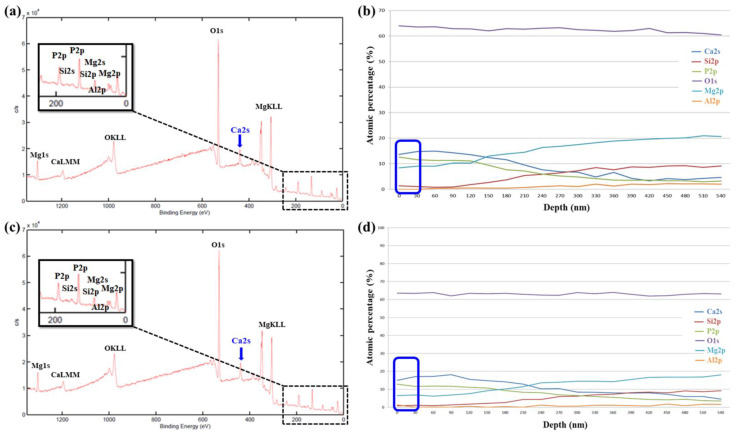
After 48 h immersion in the SBF solution, the XPS surface full spectrum and depth profile of the AZ31 sample treated in the EDTA (**a**,**b**) 0 g and (**c**,**d**) 7.5 g bath.

**Figure 11 ijms-22-04706-f011:**
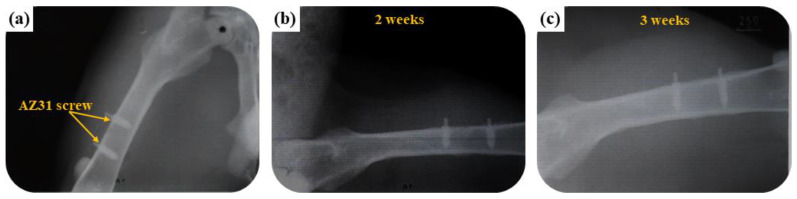
Plain X-ray of the AZ31 screw treated in the EDTA 7.5 g bath in the femoral shaft of a rabbit: (**a**) just implanted, (**b**) 2 weeks, and (**c**) 3 weeks.

**Figure 12 ijms-22-04706-f012:**
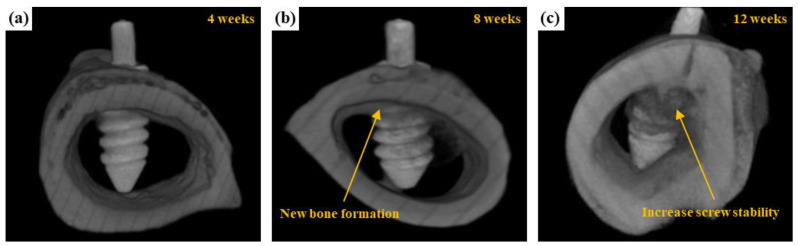
μCT reconstructed images showing the degradation processes of AZ31 bone-screw after implantation of (**a**) 4, (**b**) 8, and (**c**) 12 weeks.

**Table 1 ijms-22-04706-t001:** Electrolyte composition and operation condition for the MAO treatment.

Name	Na_2_SiO_3_	NaOH	Na_3_PO_4_	Ca_3_(PO4)_2_	EDTA
EDTA2.5	60 g/L	70 g/L	20 g/L	7.0 g/L	2.5 g/L
EDTA5.0	60 g/L	70 g/L	20 g/L	7.0 g/L	5.0 g/L
EDTA7.5	60 g/L	70 g/L	20 g/L	7.0 g/L	7.5 g/L

**Table 2 ijms-22-04706-t002:** EDS analysis of the different Ca-P containing MAO samples on AZ31 Mg alloys

	Element-Atomic%						
	C	O	Ca	Mg	Al	Si	Total
EDTA2.5	9.66	53.2	-	29.1	0.79	7.25	100
EDTA5.0	10.3	53.8	0.75	27.9	0.7	6.55	100
EDTA7.5	14.2	55.3	1.19	20.9	0.54	7.87	100

**Table 3 ijms-22-04706-t003:** Hydrogen evolution, surface roughness, and porosity of the AZ31 sample treated in the various EDTA bath

	Hydrogen Volumes (mL/14 day)	Hydrogen Evolution Corrosion Rate (mm/y)	Surface Roughness (µm)	Porosity (%)
EDTA2.5	11 ± 1.3	1.79 ± 0.38	0.85 ± 0.19	2.59 ± 0.47
EDTA5.0	9 ± 1.0	1.47 ± 0.29	0.79 ± 0.14	2.23 ± 0.45
EDTA7.5	8 ± 1.0	1.30 ± 0.21	0.62 ± 0.10	2.08 ± 0.45

**Table 4 ijms-22-04706-t004:** Locking force of the AZ31 Mg alloy bone screw

	AZ31 Mg Alloy Bone Screw
Locking force with non-immersed SBF (A)	308 ± 40 N
Locking force with immersed for 6 weeks (B)	285 ± 40 N
Residual locking force with immersed for 6 weeks (B/A)	92%
Locking force with immersed for 10 weeks (C)	265 ± 42 N
Residual locking force with immersed for 10 weeks (C/A)	86%

**Table 5 ijms-22-04706-t005:** Summary of cytotoxicity test results

Extracts	Cell Morphology	Inhibition of Viability	Cytotoxicity
Ca-P containing MAO sample	1	<30%	None
Blank control	0	<30%	None
Positive control	4	98.2%	Cytotoxicity
Negative control	0	<30%	None

**Table 6 ijms-22-04706-t006:** Content of Ca and P elements of MAO coated sample.

	Non-Immersed in SBF		After Immersed in SBF for 48 h	
	EDTA 0 g Bath	EDTA 7.5 g Bath	EDTA 0 G Bath	EDTA 7.5 g Bath
Ca	0%	3.95%	14.8%	17.1%
P	1.72%	2.08%	11.6%	11.6%

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
