# Peer review of "The Potential of Calcium/Phosphate Containing MAO Implanted in Bone Tissue Regeneration and Biological Characteristics"

_ijms, 2021, doi:10.3390/ijms22094706_

Round 1

Reviewer 1 Report

The manuscript "The Potential of Calcium/Phosphate Containing MAO Implanted in Bone Tissue Regeneration and Biological Characteristics is well written and, well structured. I recommend it for publication in its present form.

Author Response

Reviewer comments: Reviewer #1:

Comment 1:

The manuscript "The Potential of Calcium/Phosphate Containing MAO Implanted in Bone Tissue Regeneration and Biological Characteristics is well written and, well structured. I recommend it for publication in its present form.

Response:

Thank you for your appreciation. We hope that the revised version will be acceptable for publication in International Journal of Molecular Sciences.

Reviewer 2 Report

The paper entitled “ The Potential of Calcium/Phosphate Containing MAO Implanted in Bone Tissue Regeneration and Biological Characteristics" explores a micro-arc oxidation (MAO) coating produced in pulsed bipolar DC current conditions in Si- and P-containing electrolyte.

Applied on the surface of AZ31 Mg alloy it was intended to improve the biocompatibility of the test material. Various experimental techniques such as SEM, XRD, XPS, roughness measurements, electrochemical and corrosion tests, in vitro and in vivo experiments. The paper should be interesting from a scientific and practical point of view.

I would like to recommend the publication of the manuscript in this journal after fulfilling the following recommendations:

  1. More quantitative findings have to be better emphasized in the abstract;
  2. The aim of the study should be more precisely stated and better formulated.
  3. The mechanical properties of MAO coating should be evaluated not only by bone screw locking force test but with other quantitative tests such as scratch test and/or ball on wear test that could give information about the brittleness of the surface. Moreover, the bone screw locking force test was not described in section 2.
  4. The presence of amorphous Ca-P-related components is neither seen in the XRD spectra nor other tests conducted. Additional information from XPS spectra can be extracted Please, clarify this issue carefully.
  5. The biological effect of the increasing hydrogen content in time should be commented on in the text.
  6. The values shown in Tables 3 and 4 seem statistically unreliable. Standard deviation values should be added. The number of the measurements should be indicated.
  7. In Figures 8 and 9, the distribution of the chemical elements is shown at a very small scale and it could not be clearly seen. Also, for better presentation, some elements could be put to the secondary axis.
  8. It would be better for the readers to compare the animal experiments with controls that were not MAO treated.
  9. Please, pay attention to the sentences: “Applying the additives of calcium and phosphate with high concentration changes the microstructure of the coatings and produced a less porous and denser MAO structure which improves the corrosion resistance and biocompatibility. cite reference” “In this section 3.1.6 Mechanical Properties biomechanical explanation is needed with references.” and “The were no abnormality around screws in all subjects and all screws were tightly fixed to the bone in the original position”

Author Response

Reviewer comments: Reviewer #2:

Comment 1:

More quantitative findings have to be better emphasized in the abstract.

Response:

Thank you for your instruction. The correction has been made accordingly. Please see the highlights in ABSTRACT of the revised manuscript.

Comment 2:

The aim of the study should be more precisely stated and better formulated.

Response:

Thank you for your instruction. The correction has been made accordingly. Immersion process via simulated body fluid has been emphasized in the related section. Please see the highlights in INTRODUCTION of the revised manuscript.

Comment 3:

The mechanical properties of MAO coating should be evaluated not only by bone screw locking force test but with other quantitative tests such as scratch test and/or ball on wear test that could give information about the brittleness of the surface. Moreover, the bone screw locking force test was not described in section 2.

Response:

Thank you very much for the great suggestion! We are little research regarding scratch test and/or ball on wear test. And, we already planned to investigate scratches on different surfaces of MAO plated aluminum, titanium and magnesium alloys for our future work. We sincerely hope for your understanding. Additional, the information of bone screw locking force test had added in the revised manuscript. The correction has been made accordingly. Please see the highlights in SECTION 2.4 of the revised manuscript.

Comment 4:

The presence of amorphous Ca-P-related components is neither seen in the XRD spectra nor other tests conducted. Additional information from XPS spectra can be extracted Please, clarify this issue carefully.

Response:

Thank you for your instruction. We have been carefully re-checked the XRD spectra and XPS spectra. That is correct. In our experiment, Ca-P-related components are amorphous, they are not in crystalline form. So, it’s normal and natural not to be seen by XRD analysis due to its amorphous structure. However, it was needed to have additional analysis such as EDS and XPS to prove the presence of Ca and P compounds at the surface layer. After EDS and XPS results, Ca-P components were confirmed.

In our manuscript at 3.1.3. surface phase composition analysis section, we had already mentioned that:

“It is also noted that Ca and P related components were not detected in the XRD analysis. However, according to EDS analysis, Ca-P coatings were determined on the AZ31 substrates during the MAO. This is attributed to the existence of amorphous structure of Ca-P related components in the MAO treated coatings that will be revealed later in the XPS analysis.”

Comment 5:

The biological effect of the increasing hydrogen content in time should be commented on in the text.

Response:

Thank you for your instruction. “Mg alloys are attractive biodegradable medical materials, but its corrosion is accompanied by hydrogen evolution, which is detrimental to the surrounding tissue” had added in the revised manuscript. The correction has been made accordingly. Please see the highlights in page 11 of the revised manuscript.

Comment 6:

The values shown in Tables 3 and 4 seem statistically unreliable. Standard deviation values should be added. The number of the measurements should be indicated.

Response:

Thank you for your instruction. “the value of each electrochemical corrosion test, surface roughness measure, porosity analysis and mechanical test were described as the average ± of the standard deviation of the three measurements” had added in the revised manuscript. The correction has been made accordingly. Please see the highlights in page 4 and Table 3 and 4 of the revised manuscript.

Comment 7:

In Figures 8 and 9, the distribution of the chemical elements is shown at a very small scale and it could not be clearly seen. Also, for better presentation, some elements could be put to the secondary axis.

Response:

Thank you for your instruction. The correction has been made accordingly. Please see the highlights in Figure 9 and 10 (Figs. 8 and 9 of the original manuscript) of the revised manuscript.

Comment 8:

It would be better for the readers to compare the animal experiments with controls that were not MAO treated.

Response:

Thank you for your instruction. In our previous study [Ref. 30], bare AZ31 degraded quickly, whereas Ca-P free MAO screw showed a slow degradation rate. Ca-P free MAO screw maintained the original morphology within 12 weeks, whereas AZ31 screw (uncoated) suffered obvious corrosion that several pits present on the surface were observed within 4 weeks in the femoral shaft of a rabbit drill. That information had added in the revised manuscript. The correction has been made accordingly. Please see the highlights in SECTION 3.3.2 of the revised manuscript.

Comment 9:

Please, pay attention to the sentences: “Applying the additives of calcium and phosphate with high concentration changes the microstructure of the coatings and produced a less porous and denser MAO structure which improves the corrosion resistance and biocompatibility. cite reference” “In this section 3.1.6 Mechanical Properties biomechanical explanation is needed with references.” and “The were no abnormality around screws in all subjects and all screws were tightly fixed to the bone in the original position”.

Response:

Thanks for your instruction. I am very sorry for the mistake. “cite reference” and “In this section 3.1.6 Mechanical Properties biomechanical explanation is needed with references” had deleted in the revised manuscript.

This sentence “The were no abnormality around screws in all subjects and all screws were tightly fixed to the bone in the original position” misses the word “bone screws”. It has been carefully re-written in the revised manuscript. The correction has been made accordingly. Please see the highlights in page 16 of the revised manuscript.

Reviewer 3 Report

In my humble opinion, this well-written manuscript might be published as is. Just correct Ca3PO4 throughout the text. This compound does not exist. The correct formula is Ca3(PO4)2.

Author Response

Reviewer comments: Reviewer #3:

Comment 1:

In my humble opinion, this well-written manuscript might be published as is. Just correct Ca3PO4 throughout the text. This compound does not exist. The correct formula is Ca3(PO4)2.

Response:

Thanks for your instruction. I am very sorry for the mistake. The correction has been made accordingly. Please see the highlights in page 1 and 3 of the revised manuscript.

Round 2

Reviewer 2 Report

The authors have carefully addressed the reviewer's comments and recommendations.